# Identification of Surface Antigens That Define Human Pluripotent Stem Cell-Derived PRRX1+Limb-Bud-like Mesenchymal Cells

**DOI:** 10.3390/ijms23052661

**Published:** 2022-02-28

**Authors:** Daisuke Yamada, Tomoka Takao, Masahiro Nakamura, Toki Kitano, Eiji Nakata, Takeshi Takarada

**Affiliations:** 1Department of Regenerative Science, Dentistry and Pharmaceutical Sciences, Okayama University Graduate School of Medicine, Okayama 700-8558, Japan; dyamada@okayama-u.ac.jp (D.Y.); ttakao@okayama-u.ac.jp (T.T.); pptd27vr@s.okayama-u.ac.jp (T.K.); 2Precision Health, Department of Bioengineering, Graduate School of Engineering, The University of Tokyo, Tokyo 113-8656, Japan; masahiro-nakamura@umin.ac.jp; 3Department Orthopedic Surgery, Dentistry and Pharmaceutical Sciences, Okayama University Graduate School of Medicine, Okayama 700-8558, Japan; eijinakata8522@yahoo.co.jp

**Keywords:** human pluripotent stem cells, limb-bud mesenchyme, PRRX1, surface antigen

## Abstract

Stem cell-based therapies and experimental methods rely on efficient induction of human pluripotent stem cells (hPSCs). During limb development, the lateral plate mesoderm (LPM) produces limb-bud mesenchymal (LBM) cells that differentiate into osteochondroprogenitor cells and form cartilage tissues in the appendicular skeleton. Previously, we generated PRRX1-tdTomato reporter hPSCs to establish the protocol for inducing the hPSC-derived PRRX1^+^ LBM-like cells. However, surface antigens that assess the induction efficiency of hPSC-derived PRRX1^+^ LBM-like cells from LPM have not been identified. Here, we used PRRX1-tdTomato reporter hPSCs and found that high pluripotent cell density suppressed the expression of *PRRX1* mRNA and tdTomato after LBM-like induction. RNA sequencing and flow cytometry suggested that PRRX1-tdTomato^+^ LBM-like cells are defined as CD44^high^ CD140B^high^ CD49f^−^. Importantly, other hPSC lines, including four human induced pluripotent stem cell lines (414C2, 1383D2, HPS1042, HPS1043) and two human embryonic stem cell lines (SEES4, SEES7), showed the same results. Thus, an appropriate cell density of hPSCs before differentiation is a prerequisite for inducing the CD44^high^ CD140B^high^ CD49f^−^ PRRX1^+^ LBM-like cells.

## 1. Introduction

Damaged cartilage is a hallmark of osteoarthritis and rheumatoid arthritis damage cartilage. Although cartilage transplantation therapy aims to regenerate damaged cartilage [1,2], current methods, including microfracture and osteochondral autograft transplantation, are invasive and burdensome for patients.

Human cartilage tissues develop from neural crest (NC), paraxial mesoderm (PM)-derived, or lateral plate mesoderm (LPM)-derived lineages, which each exists in the cranial, axial, and appendicular skeleton, respectively [3,4]. Human pluripotent stem cells (hPSCs), including human induced pluripotent stem cells (hiPSCs) and human embryonic stem cells (hESCs), are a versatile cell resource for basic research and tissue regeneration because of their indefinite proliferation and pluripotency [5]. Several groups have induced iMSC-, NC-, or PM-derived hyaline cartilage tissues from hPSCs [6,7,8,9,10,11]. As the appendicular skeleton or limbs develop, LPM cells differentiate into paired related homeobox 1 (*PRRX1*)^+^ limb-bud mesenchymal (LBM) cells [12], and LBM-derived osteochondroprogenitors form articular cartilage and bone tissue [13]. Following ontogenic differentiation, hPSC-derived LPM cells can form LBM-like cells whose chondrogenic potential matches that of mouse LBM cells [14,15,16,17,18]. Although the clinical application of hPSC-based therapies requires highly reproducible differentiation protocols, the methods how to assess the induction efficiency of hPSC-derived LPM cells into LBM-like cells have not been established. 

Surface antigens are cell membrane-localized proteins whose expression patterns vary with cell type [19,20,21,22,23]. Identifying these surface antigens can help assess the frequency of specific cell populations and to purify them. Before chondrogenesis, the CD271^+^ CD73^−^ population should be purified from hPSC-derived NC-like cells, and then the CD271^+^ CD140A^+^ CD73^+^ CD13^−^ CDH2^−^ population can be used to test the chondrogenic capacity of NC-like-derived progenitor cells [10]. Previously, we established a protocol that not only induces almost all hPSC-derived LPM cells into PRRX1^+^ LBM-like cells but also stably expands them with a high chondrogenic capacity [14]. However, the downregulation of PRRX1 causes the loss of their chondrogenic capacity, indicating that surface antigens defining PRRX1^+^ LBM-like cells immediately prior to expansion should be identified to exclude the possible contamination of low chondrogenic cells.

In this study, we found that a high density of pluripotent cells dramatically suppresses PRRX1 expression after LBM-like induction. Using a PRRX1-tdTomato reporter hPSC line, we showed that CD44^high^ CD140B^high^ CD49f^−^ can be used to identify PRRX1-tdTomato^+^ LBM-like cells. Finally, we confirmed that PRRX1^+^ LBM-like cells derived from other hPSC lines are also CD44^high^ CD140B^high^ CD49f^−^.

## 2. Results

### 2.1. Effect of Pluripotent Cell Density before LBM-like Induction on PRRX1 Expression

Previously, we ontogenically induced PRRX1^+^ LBM-like cells from hPSCs and established their expansion method (Figure 1a) [14]. Although almost all LPM cells became PRRX1^+^ LBM-like cells following our protocol, conditions that decrease their induction efficiency have not been identified. We cultured PRRX1-tdTomato reporter hPSCs for two different time periods—three days (the condition previously used [14]) and seven days—during the pluripotent state to determine their effect on inducing PRRX1^+^ LBM-like cells. Seven-day cultures yielded higher cell densities than three-day cultures (Figure 1b). Pluripotent: 7 days showed higher cell density than Pluripotent: 3 days at Day 0 (pluripotent state). Following LBM-like cell (Day 4) induction under both conditions, *PRRX1* expression dramatically decreased after seven days of culture (Figure 1c). Pluripotent: 7 days-derived cells were called PRRX1^low^ Day 4 cells hereafter, as induction of PRRX1-tdTomato^+^ LBM-like cells was also suppressed in Pluripotent: 7 days (Figure 1d–f).

### 2.2. Identification of Surface Antigens Defining PRRX1^+^ LBM-like Cells

RNA-seq analysis of global gene expression was performed to identify surface antigens that distinguish PRRX1^+^ LBM-like cells from PRRX1^low^ Day 4 cells. Volcano plot analysis was performed to find differences in surface antigen expression between PRRX1^+^ LBM-like cells and PRRX1^low^ Day 4 cells (Figure 2a). We identified 21 genes upregulated in PRRX1+ LBM-like cells and 20 genes upregulated in PRRX1^low^ Day 4 cells that encode cell surface markers. Flow cytometry analysis showed that CD44 (HCAM) and CD140B (PDGFRB) were highly expressed in conjunction with PRRX1-tdTomato (Figure 2b,c). In contrast, CD49f (ITGA6) expression was detected only in PRRX1^low^ Day 4 cells.

### 2.3. Detection of CD44^high^ CD140B^high^ CD49f^−^ LBM-like Cells Derived from hiPSC Lines

We used four hiPSC lines, including 414C2 (the parental line of PRRX-tdTomato reporter hiPSC), 1383D2, HPS1042, and HPS1043, to analyze the expression of CD44, CD140B, and CD49f after LBM-like induction. Each cell line showed that CD44 and CD140B were highly expressed in LBM-like cells (Pluripotent: 3 days) but CD49f was only expressed in PRRX1^low^ Day 4 cells (Pluripotent: 7 days) (Figure 3a and Figure 4).

### 2.4. Detection of CD44^high^ CD140B^high^ CD49f^−^ LBM-like Cells Derived from hESC Lines

Next, we tested two hESC lines, including SEES4 and SEES7, to analyze the expression of CD44, CD140B, and CD49f after LBM-like induction. Each cell line showed that CD44 and CD140B were highly expressed in LBM-like cells (Pluripotent: 3 days), but CD49f was only expressed in PRRX1^low^ Day 4 cells (Pluripotent: 7 days) (Figure 3b and Figure 4).

## 3. Discussion

hPSC-based therapies show great potential for clinical applications, particularly cartilage regeneration. However, methods must be established to assess and increase the purity of differentiated cells to avoid teratoma formation in vivo [24]. hPSCs have high glycolytic activity, and their metabolism gradually shifts to oxidative phosphorylation during differentiation [25,26]. Several groups use metabolic shifts to kill undifferentiated hPSCs and increase the purity of differentiated cells [27,28], but surface antigens that enable us to understand their induction efficiency have not been identified. Here, we developed a surface antigen-based quality assurance to test the purity of hPSC-derived LBM-like cells. Although our previous study showed that almost all hPSC-derived LPM cells become PRRX1^+^ LBM-like cells [14], the expression of PRRX1 significantly decreased in response to the longer expansion time of hPSCs before LBM-like induction (Figure 1). Colonies of hPSCs at Pluripotent: 7 days were large and highly condensed, unlike Pluripotent: 3 days (Figure 1b). Cell density determines stem cell fate, as preculture at high cell density also changes cell sensitivity to cytokine stimulation [29,30,31,32]. Our results indicated that high cell density in the pluripotent state negatively affects the induction of PRRX1^+^ LBM-like cells from LPM cells.

RNA sequencing and flow cytometry identified hPSC-derived PRRX1^+^ LBM-like cells as CD44^high^ CD140B^high^ CD49f^−^ (Figure 2, Figure 3 and Figure 4). Freshly isolated mouse limb bud mesenchymal cells express surface antigens such as Sca1, CD105, CD90, CD73 [17]. Although the human ortholog of mouse Sca1 does not exist, we found that LBM-like cells express CD90 but do not express CD73 and CD105 (data not shown). CD44 is a hyaluronan receptor expressed by several stem and differentiated cells [33,34]. Although CD44 is highly expressed in the apical ectodermal ridge to support the proliferation of LBM cells [35], a small population of LBM cells is also CD44^+^ [17]. CD44 not only is a target gene in the WNT/beta-catenin signaling pathway but also increases its activity [36]. Our previous study showed that WNT/beta-catenin signaling upregulates PRRX1 expression, so CD44 may help LPM cells differentiate into LBM-like cells.

CD140B is a receptor for platelet-derived factors, and its surface level on Expandable LBM-like cells (ExpLBM) decreases with the downregulation of PRRX1 and chondrogenic capacity [14]. During development, LPM cells differentiate into cardiac mesoderm or LBM cells [12]. Consistent with ExpLBM, CD140B levels in LBM-like cells correlate positively with PRRX1 levels, suggesting that PRRX1 transcriptionally activates CD140B. CD49f is a marker of stem/progenitor cell populations [37], including cardiac stem cells, and was expressed here only in PRRX1^low^ Day 4 cells. High cell density inhibits WNT signaling to promote cardiomyocyte differentiation from hiPSCs [38] and the expression level of PRRX1 in cardiac mesoderm is lower than that of forelimb mesoderm [18,39], indicating that high cell density may have promoted the cardiac mesoderm-directed differentiation of LPM cells.

## 4. Materials and Methods

Cell culture. The hPSCs were cultured and maintained using StemFit (AK02N, Ajinomoto, Tokyo, Japan). Before reaching subconfluency, the cells were dissociated with TrypLE Select (Thermo Fisher, Waltham, MA, USA)/0.25 mM EDTA and suspended in StemFit containing 10 µM Y27632 (Wako, Tokyo, Japan). The cells (1 × 10^4^) were then suspended in StemFit containing 10 µM Y27632 and 8 µL iMatrix511 (human laminin-511 E8 fragment, Nippi, Tokyo, Japan) and added to a 6-cm dish. The culture medium was replaced on the next day with fresh StemFit without Y27632. The medium was changed every two days until the next passage. The following hPSC lines were used: 414C2 hiPSCs were provided by the Center for iPS Cell Research and Application, Kyoto University, Japan; 1383D2 hiPSCs, HPS1042 hiPSCs, HPS1043 hiPSCs, SEES4 hESCs, and SEES7 hESCs were donated by RIKEN BRC (Tokyo, Japan). 

Differentiation of LBM-like cells. The differentiation of hPSCs was performed as described previously [14]. In brief, hPSCs (3 × 10^4^) were suspended in 1 mL of StemFit containing 10 µM Y27632, and 4 µL of iMatrix511 were added to a 3.5-cm culture dish. The culture medium was replaced the next day with fresh StemFit without Y27632. After culturing for two or six days, the cells were differentiated to mid-primitive streak, LPM and LBM-like cells as shown in Figure 1b.

Real-time quantitative RT-PCR (qPCR). Total RNA was extracted using an RNeasy kit (QIAGEN, Germantown, MD, USA), and cDNA was synthesized using M-MLV Reverse Transcriptase (Thermo Fisher, Waltham, MA, USA) and oligo-dT primers (Sigma-Aldrich, St. Louis, MO, USA). cDNAs were then used as templates for qPCR analysis with gene-specific primers. qPCR was performed using an AriaMX Real-Time PCR System (Agilent, Santa Clara, CA, USA). The cycle parameters were as follows: denaturation at 95 °C for 30 s, annealing at 62 °C for 30 s, and elongation at 72 °C for 30 s. The expression level of each gene was calculated using the 2^−ΔΔCt^ method. The primer sequences are as follows; ACTB Forward-5′AGAAAATCTGGCACCACACC3′, ACTB Reverse-5′AGAGGCGTACAGGGATAGCA3′, PRRX1 Forward-5′TGATGCTTTTGTGCGAGAAGA3′, PRRX1 Reverse-5′AGGGAAGCGTTTTTATTGGCT3′.

Flow cytometry. Following dissociation with Accutase (Thermo Fisher, USA), 1 × 10^5^ cells were suspended in 100 µL of 2% FBS/PBS and incubated with fluorophore-conjugated antibody (200×) for 1 h on ice. The cells were then washed, and fluorescence was detected using a CytoFLEX S flow cytometer (Beckman Coulter, Brea, CA, USA). Data were analyzed using FlowJo v10.8.1 software (FlowJo LLC, Ashland, OR, USA). The antibodies used are as follows; APC-CD44 (eBioscience, Waltham, MA, USA, 17-0441), FITC-CD49f (BD, Franklin Lakes, NJ, USA, 561893), BB700-CD140B (BD, Franklin Lakes, NJ, USA, 745822).

RNA-sequence analysis. Total RNA was extracted using an RNeasy kit (Qiagen, USA), and sequencing libraries were prepared using a KAPA RNA HyperPrep Kit with RiboErase (HMR) (Kapa Biosystems, Wilmington, MA, USA) and a SeqCap Adapter Kit (Set A or Set B, Roche) following the manufacturer’s instructions. The sequencing libraries were transferred to a GENEWIZ and loaded onto a HiSeq 2500 system (Illumina, San Diego, CA, USA) for sequencing. All sequence reads were extracted in FASTQ format using the CASAVA 1.8.4 pipeline. Trimmomatic (version 0.36) was used to remove adapters and filter raw reads <36 bases as well as leading and trailing bases with < quality 20. Filtered reads were mapped to hg19 using HISAT2 software (version 2.1.0). Raw counts for each gene were based on sense-strand data obtained using featureCounts software from the Subread package. RUVSeq (release 3.10) was used for further normalization to account for sample variations. Differentially expressed genes were identified using DESeq2 analysis with a threshold of padj < 0.01 and abs (log_2_FC) > 1. Principal component analysis and a heatmap of gene expression levels of each differentially expressed marker were analyzed using the R prcomp function and hclust function (R version 3.6.1). The raw and processed RNA-seq data were deposited in the NCBI GEO database under the accession number (GSE197373).

Study approval. The Ethics Committee of Okayama University Graduate School of Medicine, Dentistry and Pharmaceutical Sciences, approved the experimental protocols for studies of human subjects. Written informed consent was provided by each donor.

Statistical analysis. Data were analyzed using Prism 9. All data were acquired by performing biological replicates of three independent experiments and are presented as the mean ± standard error of the mean. Statistical significance was determined using a two-tailed *t*-test using the Bonferroni method.

## 5. Conclusions

Our study showed that the cell density of hPSCs should be optimized before differentiation. Although human mesenchymal stromal cells (hMSCs) are widely used for cartilage regeneration, their invasive isolation from different tissues and donor-dependency inhibit stable results [40,41,42,43]. Our differentiation protocol mimics the human developmental process and provides stable results, including quality assurance of chondrogenesis, which are seldom achieved using hMSC-based current therapies. This study supports hPSC-based skeletal regeneration or disease therapy and offers novel insights on developmental biology and stem cell research.

## Figures and Tables

**Figure 1 ijms-23-02661-f001:**
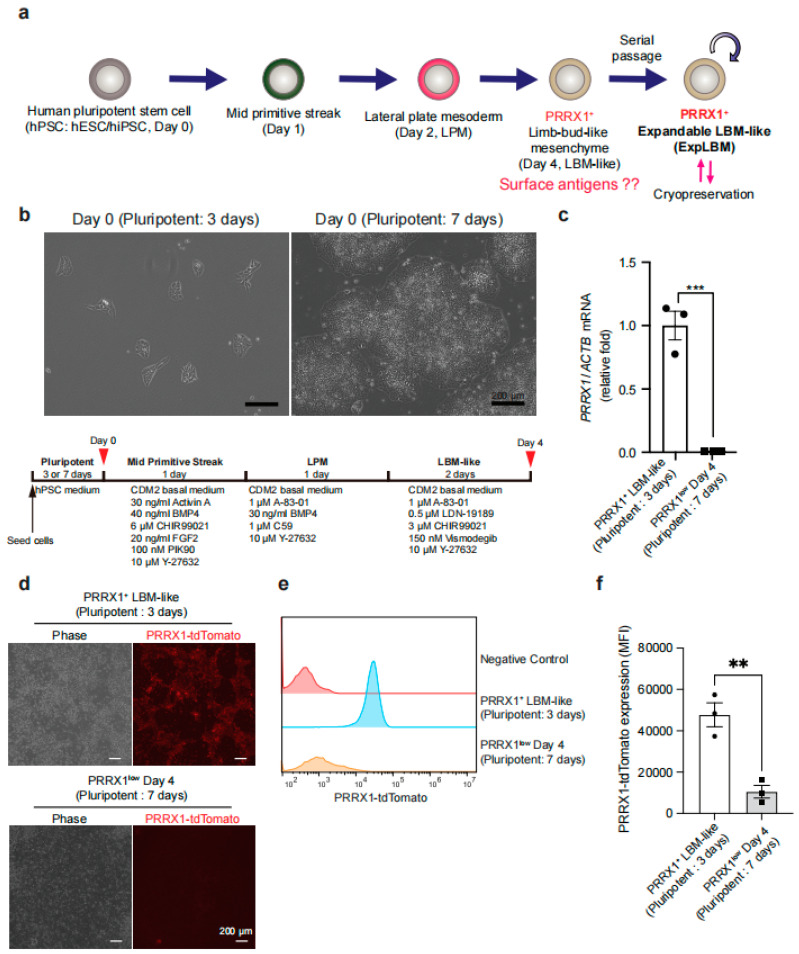
Suppressive effect of high cell density at pluripotent state on inducing PRRX1-tdTomato^+^ LBM-like cells. (**a**) Schematic overview of the induction and stable expansion of hPSC-derived LBM-like cells. ExpLBM with high chondrogenic capacity is assessed by CD90^high^CD140B^high^CD82^low^, but surface antigens that define PRRX1^+^ LBM-like cells have not been identified. (**b**) Phase images at pluripotent state just before LBM-like induction. PRRX1-tdTomato reporter hPSCs were seeded and maintained on culture dishes for 3 or 7 days. Note that cells at Pluripotent: 7 days had high cell density. (**c**) Comparison of *PRRX1* mRNA levels between Pluripotent: 3 days and Pluripotent: 7 days after LBM-like induction. Total RNA was extracted on Day 4, and *PRRX1* mRNA levels were compared using qRT-PCR. All expression values are normalized to those of *ACTB* mRNA levels (*n* = 3, three biologically independent experiments). (**d**) Images of Pluripotent: 3 days and Pluripotent: 7 days after LBM-like induction. Note that PRRX1-tdTomato was downregulated in Pluripotent: 7 days. (**e**) Flow cytometry analysis of Pluripotent: 3 days and Pluripotent: 7 days after LBM-like induction. Almost all cells derived from Pluripotent: 3 days became PRRX-tdTomato^+^, but the expression of PRRX-tdTomato was extremely low in cells derived from Pluripotent: 7 days. (**f**) Comparison of mean fluorescence intensity (MFI) values of PRRX1-tdTomato between Pluripotent: 3 days and Pluripotent: 7 days after LBM-like induction (*n* = 3, three biologically independent experiments). ** *p* < 0.01, *** *p* < 0.001.

**Figure 2 ijms-23-02661-f002:**
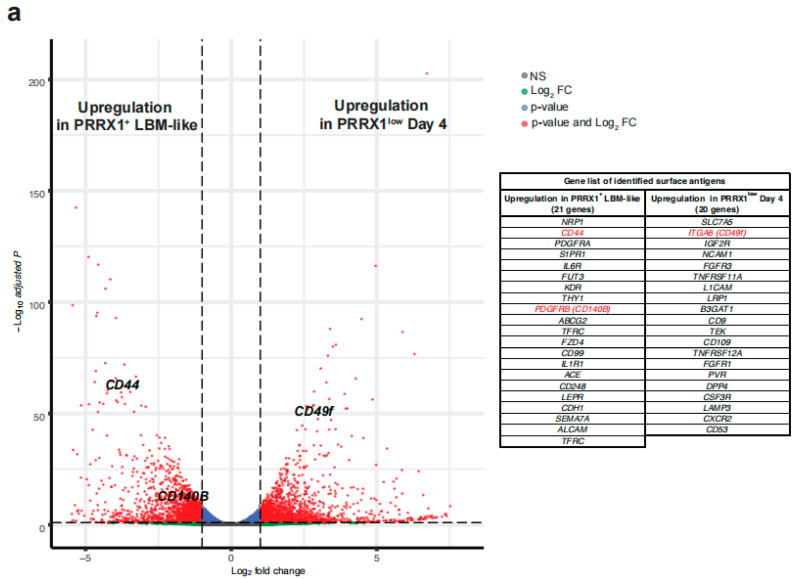
Comparison of surface antigens expressed in PRRX1-tdTomato^+^ LBM-like cells and PRRX1-tdTomato^low^ Day 4 cells. (**a**) Volcano plots to identify genes differentially expressed between PRRX1-tdTomato^+^ LBM-like cells and PRRX1-tdTomato^low^ Day 4 cells. The dashed line indicates the threshold. Red dots represent differentially expressed genes (*n* = 2, two biologically independent experiments, padj < 0.01, log_2_FC > 1). (**b**) Representative flow cytometry analysis data for CD44 (HCAM), CD140B (PDGFRB), and CD49f (ITGA6) expressed by PRRX1-tdTomato^+^ LBM-like cells or PRRX1-tdTomato^low^ Day 4 cells. (**c**) Comparison of mean fluorescence intensity (MFI) values of CD44, CD140B or CD49f between Pluripotent: 3 days and Pluripotent: 7 days after LBM-like induction (*n* = 3, three biologically independent experiments). * *p* < 0.05, *** *p* < 0.001.

**Figure 3 ijms-23-02661-f003:**
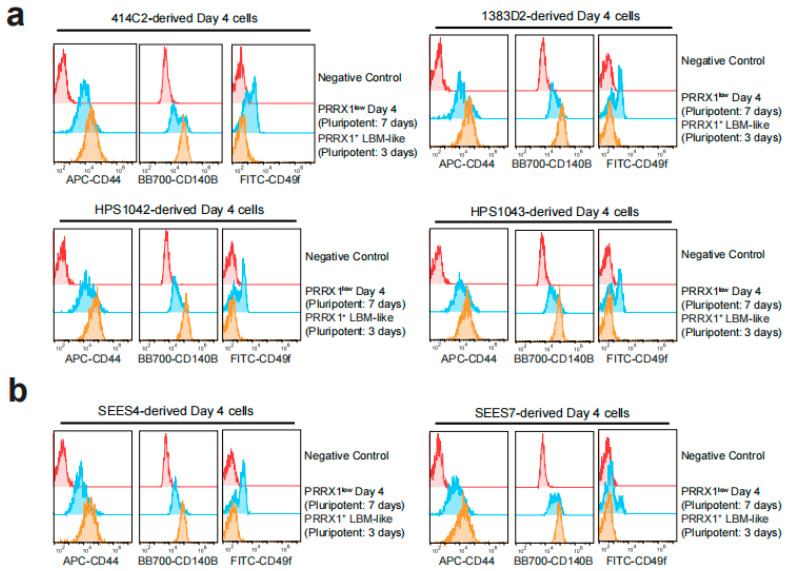
Flow cytometry analysis for comparing the surface levels of CD44, CD140B, and CD49f between hiPSC or hESC-derived PRRX1^+^ LBM-like cells and PRRX1^low^ Day 4 cells. (**a**) Representative flow cytometry analysis data for CD44 (HCAM), CD140B (PDGFRB), and CD49f (ITGA6) expressed by hiPSC (414C2, 1383D2, HPS1042, and HPS1043)-derived LBM-like cells or PRRX1^low^ Day 4 cells. (**b**) Representative flow cytometry analysis data for CD44 (HCAM), CD140B (PDGFRB), and CD49f (ITGA6) expressed by hESC (SEES4 and SEES7)-derived LBM-like cells or PRRX1^low^ Day 4 cells.

**Figure 4 ijms-23-02661-f004:**
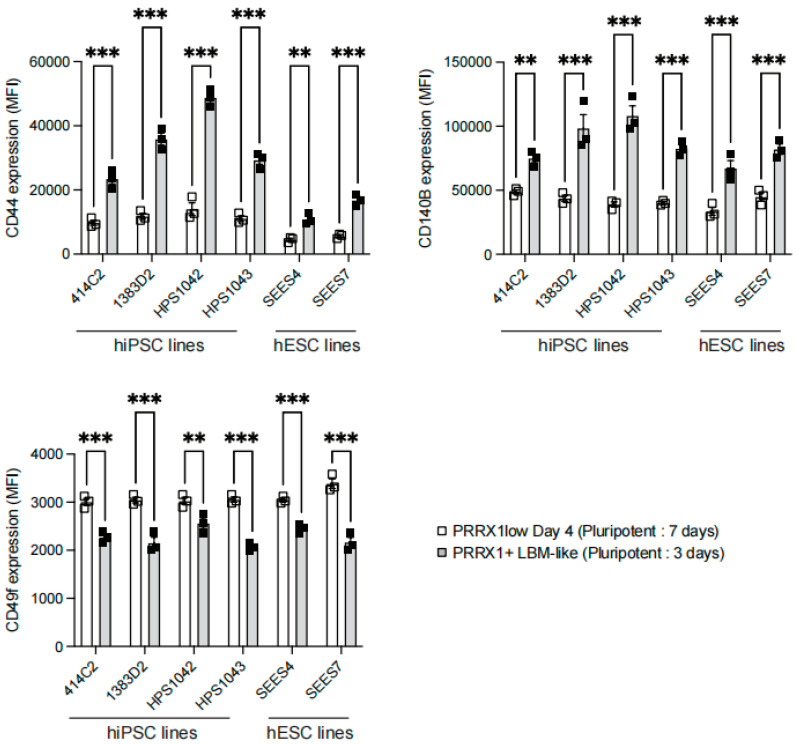
Quantification of the surface levels of CD44, CD140B, and CD49f among hiPSC or hESC-derived PRRX1^+^ LBM-like cells and PRRX1^low^ Day 4 cells. Comparison of mean fluorescence intensity (MFI) values of CD44, CD140B or CD49f expressed by hiPSC (414C2, 1383D2, HPS1042, and HPS1043) or hESC (SEES4 and SEES7)-derived LBM-like cells or PRRX1^low^ Day 4 cells (*n* = 3, three biologically independent experiments). ** *p* < 0.01, *** *p* < 0.001.

## Data Availability

All data generated during and/or analyzed during this study are included in this published article.

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
