# Peer review of "Identification of Surface Antigens That Define Human Pluripotent Stem Cell-Derived PRRX1+Limb-Bud-like Mesenchymal Cells"

_ijms, 2022, doi:10.3390/ijms23052661_

Round 1

Reviewer 1 Report

I read Your article with great interest as cartilage damage is a great problem without an adequate treatment in many cases. I found the study design clear and adecvate and the results are also in concordance with the main hypothesis. 

I endorse the manuscript publication as it is. 

Author Response

Response: We thank for this comment, but our manuscript was revised according to other reviewer`s comments.

Reviewer 2 Report

This study is aimed to identify the surface antigen that could be a useful tool to define human pluripotent stem cell(hPSCs)- derived PRRX1+ limb-bud like mesenchymal cells using PRRX1-tdTomato reporter hPSCs. It would be an informative finding for further studies that apply these cells detection or isolation. 

However, there are several issues need to be addressed: 

Major

  1. In statistical analysis description, it is mentioned that “All data were acquired by performing biological replicates of two or three independent experiments”. How can they get statistical significant different if their n samples less than 3 replications?
  2. For the flow cytometry data in figures 1-4, they should show the quantification data (for example, MFI value?)
  3. They mentioned that they identified 21 genes and 20 genes upregulated in PRRX1+ LBM like cells and PRRX1low day 4 cells, respectively. Could they list those genes? Is there any genes downregulated?
  4. It could be more informative if they show direct comparison the data quantification between Figure 3 and 4?
  5. It’s little bit confusing, whether “PRRX1low day 4 cells” are considered as ExpLBM or not? if not how we can distinguish LBM-like from ExpLBM-like cells? This should be also discussed in this article.
  6. Additional discussion regarding whether other limb bud mesenchymal cells surface markers that usually are found in freshly isolated limb bud mesenchymal cells, such as Sca1, CD105, CD90, CD73 (ref 17) are also observed in PRRX1+ LBM like cells or not could make discussion section more comprehensive.

Minor

  1. In figure 1 b and c, they show pluripotent: 3 days before 7 days, but in figure 1 d they put reverse figure order? It would be easier to read if they have consistency in arranging the data.
  2. Figure 2a not in the high resolution, it’s hard to read the legend

Author Response

Major

  1. In statistical analysis description, it is mentioned that “All data were acquired by performing biological replicates of two or three independent experiments”. How can they get statistical significant different if their n samples less than 3 replications?

Response: We thank for this comment. We corrected the description (page 5, line 21)

  1. For the flow cytometry data in figures 1-4, they should show the quantification data (for example, MFI value?)

Response: We agree with this comment. The MFI value of each flowcytometry data was added and manuscript was also revised.

  1. They mentioned that they identified 21 genes and 20 genes upregulated in PRRX1+ LBM like cells and PRRX1low day 4 cells, respectively. Could they list those genes? Is there any genes downregulated?

Response: As mentioned by this comment, the gene list should be shown in the manuscript. We added the gene list in Figure 2a. Our description in manuscript means that compared with PRRX1+ LBM-like cells, mRNA levels of 21 and 20 cell surface marker genes in PRRX1low Day 4 cells were downregulated and upregulated, respectively.

  1. It could be more informative if they show direct comparison the data quantification between Figure 3 and 4?

Response: To respond to this comment, we rearranged Figure 3 and 4 as follows: Figure 3 - Flowcytometry data of hiPSC or hESC-derived cells, Figure 4 - Comparison of mean fluorescence intensity (MFI) values of CD44, CD140B or CD49f expressed by hiPSC (414C2, 1383D2, HPS1042, and HPS1043) or hESC (SEES4 and SEES7)-derived cells.

  1. It’s little bit confusing, whether “PRRX1low day 4 cells” are considered as ExpLBM or not? if not how we can distinguish LBM-like from ExpLBM-like cells? This should be also discussed in this article.

Response: As previously demonstrated in our paper (ref 14), the serial passage of PRRX1+LBM-like cells (Day 4 cells) induces more in vivo LBM-like gene expression profile to produce ExpLBM. In other words, ExpLBM are PRRX1+LBM-like cells that experienced serial passage. Since our descriptions about LBM-like and ExpLBM in this manuscript may make the readers to confuse, we revised our manuscript.

  1. Additional discussion regarding whether other limb bud mesenchymal cells surface markers that usually are found in freshly isolated limb bud mesenchymal cells, such as Sca1, CD105, CD90, CD73 (ref 17) are also observed in PRRX1+ LBM like cells or not could make discussion section more comprehensive.

Response: Although human ortholog of mouse Sca1 do not exist, we found that LBM-like cells express CD90 but do not express CD73 and CD105. These descriptions were added in discussion section (page 3, line 28-31).

Minor

  1. In figure 1 b and c, they show pluripotent: 3 days before 7 days, but in figure 1 d they put reverse figure order? It would be easier to read if they have consistency in arranging the data.

Response: We agree with this comment and rearranged figure 1d.

  1. Figure 2a not in the high resolution, it’s hard to read the legend

Response: We improved the resolution of Figure 2a and its legend became clear.

Reviewer 3 Report

The authors aimed to developed a surface antigen-based quality assurance to test the purity of hPSC-derived LBM-like cells. The study covers some issues that have been overlooked in other similar topics. The structure of the manuscript appears adequate and well divided in the sections. Moreover, the study is easy to follow, but few issues should be improved. Some of the comments that would improve the overall quality of the study are:

1-) Abstract and Introduction section: Please better describe the aim of the work;

2-) The manuscript needs grammar correction. Please also check typos thorough the text;

3-) Limitations of the study needs to better addressed;

4-) Conclusion Section: This paragraph is missing. Please add it.

Author Response

1-) Abstract and Introduction section: Please better describe the aim of the work;

Response: We agree with this comment and the descriptions about the aim of this work were added in our manuscript (page 1, line 21-24 / page 2, line 2-4 / page 2, line 13-16).

2-) The manuscript needs grammar correction. Please also check typos thorough the text;

Response: We thank for this comment and our manuscript was corrected by English editing service.

3-) Limitations of the study needs to better addressed;

Response: To make our study more informative, we added the MFI value of each cytometry data to Figures and revised manuscript.

4-) Conclusion Section: This paragraph is missing. Please add it.

Response: We thank for this comment and Conclusion Section was added followed by Discussion section (page 4, line 1-9).

Round 2

Reviewer 2 Report

The authors have made significant improvement to the manuscript.

Minor suggestions:

  1. Please check the figure format and consistency in the text font (for example reference section)
  2. Format for each material /reagent in method should has brand, purchase from (for example: BD Biosciences, San Jose, CA, USA)
  3. Figure 4, the symbol size for data point is too large, should follow figure 2C

Author Response

  1. Please check the figure format and consistency in the text font (for example reference section)

Response: We thank you for this comment and text font was changed to the same.

  1. Format for each material /reagent in method should has brand, purchase from (for example: BD Biosciences, San Jose, CA, USA)

Response: We thank for this comment and country names were added to Materials & Methods section.

  1. Figure 4, the symbol size for data point is too large, should follow figure 2C

Response:  We thank for this comment and made the symbol size of each data point smaller.